# Initial Stage of Formation of Spontaneous Invasive Populations of Garden Lupine (*Lupinus polyphyllus* Lindl.) at the Northern Limit of Its Secondary Distribution Range in the Veps Forest Nature Park

**Maria A. Galkina** [1,*], **Yulija K. Vinogradova** [1], **Viktoria N. Zelenkova** [2], **Natalia V. Vasilyeva** [1], **Ekaterina V. Tkacheva** [3] **and Olga V. Shelepova** [1,4,*] 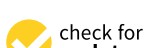

[1] N.V. Tsitsin Main Botanical Garden of Russian Academy of Sciences, Moscow 127276, Russia
[2] Botanical Garden BNSU, Belgorod National State University, Belgorod 308015, Russia
[3] Library for Natural Sciences of Russian Academy of Sciences, Moscow 119991, Russia
[4] All-Russia Research Institute of Agricultural Biotechnology, Timiryazevskaya St., 42, Moscow 127550, Russia
*   Correspondence: mawa.galkina@gmail.com (M.A.G.); shov_gbsad@mail.ru (O.V.S.)

**Abstract:** A weed plant of the species *Lupinus polyphyllus* was found in settlements located on the territory of the natural park "Vepssky forest" of the Leningrad region. The plant is known as a transforming species in the southern regions of Russia. The distribution of *L. polyphyllus* within the Vepsian forest occurs not due to the migration of the species to the north along transport routes, as one might assume, but due to flight from gardens and the formation of spontaneous invasive populations. The goal of the study was to identify the molecular and biochemical characteristics of garden lupine at the northern borders of its secondary range. To interpret the obtained data, the task of the study also included a comparison of intraspecific variability between the "old" invasive populations (in Finland and Central Russia, studied by us earlier) and the "new" naturalizing population of *L. polyphyllus* in the Vepsian forest. The search for *L. polyphyllus* localities in the territory of the Vepssky Les natural park was carried out by the route method with geobotanical descriptions of experimental sites (5 m × 5 m). DNA was isolated from eight herbarium specimens of *L. polyphyllus* (MHA, LE) and fifteen specimens collected in the territory of the natural park "Vepssky Les". To reveal the internal structure and phylogenetic relationships in lupine populations, networks of nuclear and chloroplast haplotypes and cluster analysis (UPGMA) with the SplitsTree program were used. The total content of polyphenols and flavonoids in the leaves was determined spectrophotometrically. The low inter-locality variability of ITS indicates that the populations of *L. polyphyllus* in Central Russia and in the North of Russia (St. Petersburg and Vepsskaya Pushcha) have the same origin. Analysis of the chloroplast intergenic spacer (rpl32–trnL) indicates intrapopulation diversity and suggests the presence of microevolutionary processes near the northern limits of the secondary distribution range of *L. polyphyllus.* The high content of polyphenols and flavonoids in the leaves reveals the adaptive capabilities of lupine in the studied area. Evidence suggests that a neglected species may soon become invasive, as has already happened in other regions.

**Keywords:** weeds; biological invasions; secondary distribution range; *Lupinus polyphyllus*; ITS; rpl32–trnL; polyphenols; flavonoids; polymorphism; microevolution

## 1. Introduction

Due to the growing anthropogenic pressure, the rate of formation of semi-natural habitats has been ever-increasing, inevitably leading to disturbances of the vegetation cover and a decrease of biodiversity. In addition to the disappearance of stenotopic species, an increase of weedy species occurs here. Weeds are often alien species that constitute a major threat to the native flora [1]. Weedy species may be capable of spreading from the disturbed

habitats into nearby natural plant communities, where they are a threat to native and, most notably, rare and protected species, as well as into agroecosystems, where they cause losses of crops.

Moreover, following rapid climate change within the past decades, a shift of range limits can be observed in many species [2]. Global warming leads the expansion of species ranges to be most evident in higher latitudes [3]. The rate of plant invasions has also been increasing [4–6]. The expansion of alien species is now a global trend and has caused drastic changes within native ecosystems, most importantly, negatively affecting biodiversity on all its levels, from populations to ecosystems [7–9]. Invasive species constitute a major threat to indigenous communities, such as boreal taiga forests, which form a substantial part of the vegetation cover within the Veps Forest Nature Park. Moreover, it has been demonstrated that communities already affected by invasive species become more vulnerable to further invasions [10].

Veps Forest Nature Park, established in 1999, is a protected area of regional significance [11]. The nature park is located in the northern part of the Onega-Valdai Hills, and most of its territory is covered with bilberry spruce forests [11]. The protected area was later expanded to include several natural sanctuaries and their surroundings with limited anthropogenic activity, a territory with several rural settlements. Despite limited anthropogenic activity within the protected area, numerous plant species are cultivated on residential yards and adjacent allotments of various organizations.

Over the last few years, the secondary range of *Lupinus polyphyllus* Lindl. within Leningrad Oblast has been expanding [12]. *L. polyphyllus* (Fabaceae) is an herbaceous biennial or short-lived perennial of North American origin. The native range of the garden lupine lies within western North America: Canada (British Columbia) and the USA (Alaska, Washington, Oregon, and California) [13]. In 1826, *L. polyphyllus* Lindl. Was brought to England by the famous Scottish "plant hunter" David Douglas and soon became widely cultivated in Central Europe as an ornamental plant [14]. As early as the 1840s, nurseries and botanical gardens offered some color variations of this species. They were described in the rank of forms: *L. polyphyllus* f. *roseus* Voss, *L. polyphyllus* f. *tricolor* Voss, *L. polyphyllus* f. *albus* (Regel) Voss, and *L. polyphyllus* f. *atropurpureus* Voss [15]. From 1935–1937, the garden hybrid *L.* × *regalis* Bergmans = *L. agboreus* Sims. × *L. polyphyllus*, which is characterized by a branched stem with dense inflorescences, blue, purple, pink, white, and even yellowish or orange colorations were recorded. Some articles attribute wild European populations of *L. polyphyllus* with a variety of flower colors to this hybrid. However, our previous studies have shown that there is no reason for this [16].

The species was first noted in the list of "escapees" from a garden in Sweden in 1870 [17]. In Germany, wild populations have been found since 1890 and in Finland since 1895 [18]. The introduction of lupine into natural cenoses of Northern Europe was mainly due to cultivation for soil substrate stabilization [19–21]. The species is currently listed as invasive in 15 European countries.

In Central Russia in the second half of the 20th century, this species was widely cultivated as a green fertilizer, so its pathway of invasion is "escaping" not from a garden, but from an agricultural crop. Invasive populations with polymorphic flower colorations have been observed since 1974 [16]. *L. polyphyllus* is a cross-pollinated entomophilous species, but self-pollination is also not excluded. The species is characterized by high seed productivity. In the Smolensk Region, 5-year-old individuals produce 1.3 t/ha of seeds [22]. Seed germination persists for >50 years [23]. *L. polyphyllus* is capable of vegetative propagation by particulation of the caudex and dividing the plant into several daughter clones. However, these clones are genetically old and do not always flower and bear fruit. Currently, the species is widely distributed in Russia and has been included in the list of the most aggressive invasive species in Russia (Top 100) [24]. *L. polyphyllus* in Central Russia actively invades natural plant communities, equally intruding into meadows and forests with varying crown density [25]. The study of intraspecific variability is extremely relevant for the development of population control measures for this species. In most habitats within

Central Russia, it has become impossible to trace the origin of specific lupine populations. However, in the Veps Forest, *L. polyphyllus* was found only in disturbed plant communities within settlements where it escaped from residential yards and neglected allotments of a museum and nearby schools. These initial populations are of peculiar interest; with the use of molecular genetic methods, their origin and vector of invasion may be identified quite accurately.

Furthermore, the adaptation of plants to new growth conditions is strongly linked with the regulation of the synthesis of phenolic compounds, as these particular molecules often exert a long-term influence on the growth and survival rate under new stress conditions [26]. The survival, longevity, productivity, and invasiveness of plants depend on increased synthesis of secondary metabolites, and in particular, on the synthesis of a number of phenolic compounds and flavonoids that provide protective responses against abiotic stress [27]. Thus, lower temperatures enhance the synthesis of phenolic compounds and subsequent integration of these compounds into the plant cell wall in the form of lignin, which allows plants to adapt to cold conditions [28]. In addition, increased UV radiation due to changes in the ozone layer contributes to oxidative stress, which in turn increases the biosynthesis of flavonoids. These compounds act as antioxidants and protect plants from the effects of oxidative stress [29]. Therefore, the fluctuation of phenolic content makes it possible to estimate the adaptation success rate of *L. polyphyllus* under new growth conditions.

It is known that the "founder effect" plays a determining role in the formation of the species' secondary range. The initial stage of naturalization of the species from a few (or even one) initial diaspora is observed in the Veps forest. Therefore, it was interesting to compare the intrapopulation variability of this initial invasive population at the northern border of the species' range with the variability of invasive populations in Finland and in Central Russia that we studied earlier [16].

The aim of the study was to evaluate the invasion activity of *L. polyphyllus* within the Veps Forest Nature Park and to determine its molecular-genetic and biochemical characteristics initiated by microevolution processes over the course of the expansion of its secondary distribution range. To interpret the data obtained, the aim of the study also includes a comparison of intraspecific variability between the "old" invasive populations and the "new" naturalizing population of *L. polyphyllus*.

## 2. Materials and Methods

### 2.1. Field Research

The search for *L. polyphyllus* occurrences in the territory of the Veps Forest Nature Park (Figure 1) was carried out using a route-based method. Routes were laid in such a way as to cover the maximum diversity of habitats, and within each of them, there was a segment of the route of maximum length. Due to the biological features of *L. polyphyllus*, we did not include bogs in the routes. When moving along the riverbed of the Oyat River, both banks were investigated. In sites where garden lupine was located, we conducted geobotanical descriptions (sample plot size 5 m × 5 m) using the classic method [30]. Species nomenclature follows the World Flora Online (worldfloraonline.org accessed on 16 September 2022) database.

### 2.2. Molecular Data

DNA was extracted from herbarium specimens of *L. polyphyllus* housed at the MHA and LE herbaria and from specimens collected within the Veps Forest Nature Park (Tables 1 and S1) and dried using silica gel. The sample consisted of 23 specimens (15 specimens from four lupine populations in the Veps Forest, 8 specimens from the herbaria).

DNA was extracted using the Diamond DNA Kit (OOO "AltayBioTech") and amplified using Biorad T-100 (USA). Primers used for PCR were synthesized and purified in PAAG by Syntol Ltd. (Moscow, Russia). Polymerase chain reaction (PCR) was performed in a total volume of 20 μL, containing 4 μL of Ready-to-Use PCR MasterMIX based on "hot-

start" SmarTaq DNA Polymerase (Dialat Ltd., Moscow, Russia), 13 μL of deionized water, 3.2 pmol of each primer, and about 1.5–2 ng of template DNA. For the nuclear ribosomal internal transcribed spacer 1–2 (ITS1–2), the primers nnc18s10 (forward) and c26A (reverse) were used with an annealing temperature of 58 °C. For chloroplast intergenic non-coding spacer rpl32–trnL, primers rpl32F (forward) and trnL UAG (reverse) were used with an annealing temperature of 57 °C. Purification of the PCR product for sequencing was carried out in a mixture of ammonium acetate with ethanol. The nucleotide DNA sequences were determined on an automatic sequencer (Syntol). The data were submitted to GenBank [31], in which these nucleotide sequences can be found by their accession numbers (Table S1).

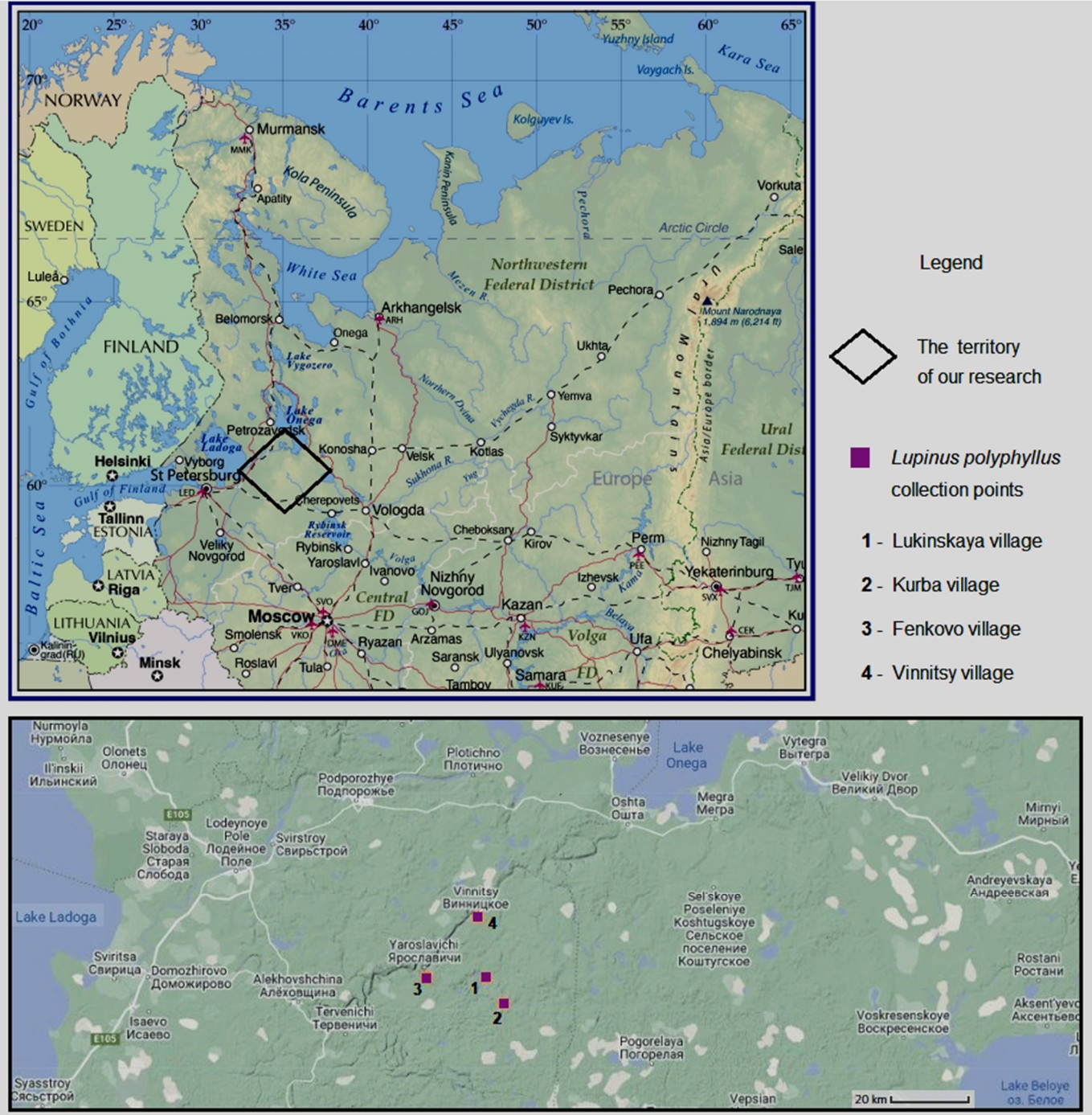

**Figure 1.** Sample locations in the Veps Forest Nature Park.

**Table 1.** Samples of *Lupinus polyphyllus* locations.

| No. Sample | Place of Collection | No. Sample | Place of Collection |
| --- | --- | --- | --- |
| LP1 | Saint-Petersburg | LS1 | Smolensk Oblast, Demidov district, near Borovichi village. 55°17′25″ N, E31°29′46″ E |
| LP2 | Leningrad Oblast | LV1a, LV1b, LV1c, LV1d, LV1e | Leningrad Oblast, Podporozhye district, Veps Forest Nature Park, Lukinskaya village. 60°27′35″ N, 34°50′04″ E |
| LT1, LT2 | Tula, Lazo street. 51°09′50″ N, 37°36′32″ E | LV2a, LV2b, LV2c, LV2d, LV2e | Leningrad Oblast, Podporozhye district, Veps Forest Nature Park, Kurba village. 60°23′58″ N, 34°56′03″ E |
| LK1 | Kaluga Oblast, Yukhnov district, near Gorodets village. 54°27′43″ N, 35°01′30″ E | LV3 | Leningrad Oblast, Podporozhye district, Veps Forest Nature Park, Fenkovo village. 60°28′39″ N, 34°29′51″ E |
| LK2a, LK2b | Kaluga Oblast, Zhukov district, Olkhovo village. 55°11′20″ N, 36°57′36″ E | LV4a, LV4b, LV4c, LV4d | Leningrad Oblast, Podporozhye district, Veps Forest Nature Park, Vinnitsy village. 60°37′44″ N, 34°45′47″ E |

DNA was extracted using the Diamond DNA Kit (OOO "AltayBioTech") and amplified using Biorad T-100 (USA). Primers used for PCR were synthesized and purified in PAAG by Syntol Ltd. (Moscow, Russia). Polymerase chain reaction (PCR) was performed in a total volume of 20 µL, containing 4 µL of Ready-to-Use PCR MasterMIX based on "hot-start" SmarTaq DNA Polymerase (Dialat Ltd., Moscow, Russia), 13 µL of deionized water, 3.2 pmol of each primer, and about 1.5–2 ng of template DNA. For the nuclear ribosomal internal transcribed spacer 1–2 (ITS1–2), the primers nnc18s10 (forward) and c26A (reverse) were used with an annealing temperature of 58 °C. For chloroplast intergenic non-coding spacer rpl32–trnL, primers rpl32F (forward) and trnL UAG (reverse) were used with an annealing temperature of 57 °C. Purification of the PCR product for sequencing was carried out in a mixture of ammonium acetate with ethanol. The nucleotide DNA sequences were determined on an automatic sequencer (Syntol). The data were submitted to GenBank [31], in which these nucleotide sequences can be found by their accession numbers (Table S1).

*2.3. Determination of Total Polyphenol and Flavonoid Content*

The total content of polyphenols and flavonoids was determined in the leaves of the plant. We sampled from each specimen the fourth upper leaf of a lupine shoot. Samples were collected from each of the studied populations in the Veps Forest. The material was collected in early summer (10–12 June); the leaves were fully unfolded and were not damaged by pests or drought.

The total polyphenol content was measured by the method of [32] using the Folin–Ciocalteu reagent. A quantity of 0.075 cm$^3$ of each sample was mixed with 0.075 cm$^3$ of the Folin–Ciocalteu reagent diluted 5-fold after 3 min 0.15 cm$^3$ of 20% (*w/v*) sodium carbonate and 1.2 cm$^3$ of distilled water. After 60 min in darkness, the absorbance at 725 nm was measured with the spectrophotometer Spekol 1300 (Analitik Jena AG, Jena, Germany). Gallic acid (25–300 mg/L; $R^2$ = 0.998) was used as the standard. The results were expressed in mg/g DM gallic acid equivalent.

The total flavonoid content was determined using the modified method described by [33]. An aliquot of 1 cm$^3$ of the sample was mixed with 2 cm$^3$ of 2% (*w/v*) ethanolic solution of aluminum chloride, 0.5 cm$^3$ of 1 M hydrochloric acid, and 6.5 cm$^3$ of ethanol (96%). After 30 min in darkness, the absorbance at 415 nm was measured using the spectrophotometer Spekol 1300 (Analitik Jena AG, Jena, Germany). Quercetin (1–400 mg/L; R$^2$ = 0.9977) was used as the standard. The results were expressed in mg/g DM quercetin equivalent.

### 2.4. Analysis of Data

Sequences were checked and manually edited and aligned using BioEdit v. 7.0.5.3. program [34]. All alignments were built from consensus sequences obtained by direct sequencing of PCR products. We paid special attention to careful examination of electropherograms to identify sites with nucleotide substitutions. Evolutionary analysis for chloroplast intergenic spacer rpl32–trnL was conducted in SplitsTree [35]. The tree was obtained automatically by applying UPGMA algorithms. Branch support was assessed with 1000 bootstrap replications. The phylogenetic tree was drawn to scale, with branch lengths measured in the number of substitutions per site. We constructed the haplotype networks for ITS site and rpl32–trnL using TCS 1.21 program [36].

### 3. Results and Discussion

The invasion of garden lupine within the Veps Forest Nature Park is currently in its initial stage. The plants were located in neglected allotments within several dozens of meters from the nearest schools (Lukinskaya and Vinnitsy villages) and a museum (Kurba village); one specimen was found in Fenkovo village within a few meters from a residential yard (Figure 2). The cover of lupine varied from 1 to 30% (Table 2), which is a rather low level. It has been recorded that in Central Russia, as well as in higher latitudes in Finland, lupine is capable of forming monodominant stands [8,37].

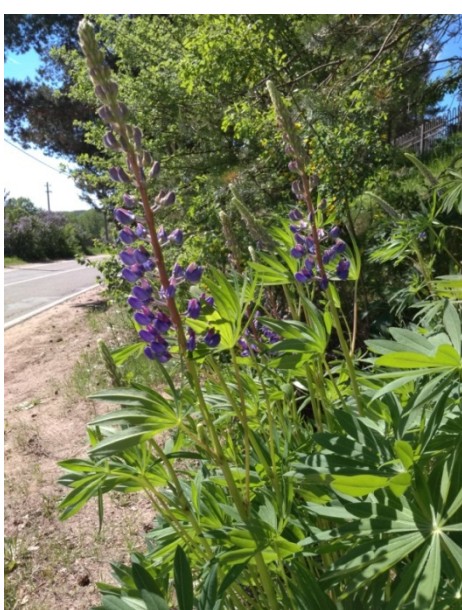

**Figure 2.** *Lupinus polyphyllus* on the side of the road in the village of Vinnitsy.

All of the studied plant communities (Table 2) were located along dirt roads within settlements and suffered major anthropogenic disturbances. These were not climax communities of the northern taiga but secondary birch forests, sometimes with pine. The herbaceous layer was dominated by weedy and ruderal species (*Aegopodium podagraria*, *Trifolium pratense*, *Urtica dioica*, etc.) rather than forest species. Moreover, several alien species had already invaded these communities, namely: *Solidago canadensis* L., widely

distributed in Russia, and *Veronica filiformis* Sm., actively spreading in the latest decade. On the sample plots, 13 to 20 vascular plant species were found which belonged to 18 families, most notably Rosaceae, Fabaceae, Asteraceae, and Plantaginaceae (Table 2). Lupine was not found in non-disturbed meadow or forest habitats within the Veps Forest. Therefore, the invasion of lupine was, with certainty, observed at its initial stage; lupine may be considered a species with the invasive status 3, i.e., an alien species presently expanding and becoming naturalized in disturbed habitats, which, following further naturalization, may become capable of entering semi-natural and natural habitats [38].

**Table 2.** Geobotanical descriptions of plant communities with *L. polyphyllus*.

| Location | Lukinskaya Village 60°27′35″ N 34°50′04″ E | Kurba Village 60°23′58″ N 34°56′03″ E | Fenkovo Village 60°28′39″ N 34°29′51″ E | Vinnitsy Village 60°37′44″ N 34°45′47″ E |
|---|---|---|---|---|
| Species | Cover (%), for Trees–Crown Density (%) | | | |
| A. Tree tier | 10 | 15 | | 15 |
| *Betula alba* L. | 15 | 15 | | 6 |
| *Pinus sylvestris* L. | | | | 9 |
| B. Schrubs and undergrowth | 5 | 10 | | 15 |
| *Betula alba* L. | 2 | | | 5 |
| *Pinus sylvestris* L. | | | | 5 |
| *Prunus padus* L. | 2 | | | |
| *Populus tremula* L. | 2 | | | |
| *Ribes uva-crispa* L. | | <1 | | |
| *Rosa majalis* L. | | <1 | | 2 |
| *Sorbus aucuparia* L. | | <1 | | |
| C. Grass-shrub tier | 90 | 70 | 90 | 50 |
| *Achillea millefolium* L. | 1 | <1 | 2 | 1 |
| *Aegopodium podagraria* L. | | 1 | 25 | |
| *Alchemilla vulgaris* L. | | 10 | 5 | <1 |
| *Angelica sylvestris* L. | | | 5 | |
| *Anthriscus sylvestris* (L.) Hoffm. | 20 | 2 | 20 | |
| *Artemisia vulgaris* L. | | | | 1 |
| *Campanula trachelium* L. | 5 | | | |
| *Dactylis glomerata* L. | 20 | 1 | 15 | |
| *Elytrigia repens* (L.) Nevski | | | | <1 |
| *Equisetum arvense* L. | | 1 | | |
| *Fragaria vesca* L. | | 7 | | 1 |
| *Gentiana cruciata* L. | 1 | | | |
| *Geranium sylvaticum* L. | 1 | 2 | | |
| *Hieracium umbellatum* L. | | | | 1 |
| *Iris pumila* L. | 10 | | | |
| *Knautia arvensis* (L.) Coult. | | | | <1 |

**Table 2.** *Cont.*

| Location | Lukinskaya Village 60°27′35″ N 34°50′04″ E | Kurba Village 60°23′58″ N 34°56′03″ E | Fenkovo Village 60°28′39″ N 34°29′51″ E | Vinnitsy Village 60°37′44″ N 34°45′47″ E |
|---|---|---|---|---|
| *Lupinus polyphyllus* Lindl. | 15 | 20 | 1 | 30 |
| *Plantago lanceolata* L. | | | | <1 |
| *Plantago major* L. | | | 1 | |
| *Poa annua* L. | | | 1 | |
| *Poa compressa* L. | | 25 | | |
| *Poa palustris* L. | | | | 1 |
| *Poa pratensis* L. | 2 | | | |
| *Ranunculus repens* L. | 1 | 1 | 2 | |
| *Rumex confertus* Willd. | | | | <1 |
| *Seseli libanotis* (L.) W.D.J. Koch | 1 | | | 1 |
| *Solidago canadensis* L. | 2 | | | |
| *Taraxacum campylodes* G.E. Haglund | 5 | | 2 | |
| *Trifolium pratense* L. | | 10 | 10 | |
| *Urtica dioica* L. | 1 | | | |
| *Veronica chamaedrys* L. | 10 | 25 | 20 | 5 |
| *Veronica filiformis* Sm. | | | | 1 |
| *Veronica serpyllifolia* L. | | 1 | | |
| *Vicia cracca* L. | 2 | 2 | | 10 |
| D. Moss-lichen tier | | 5 | | |

The main vector of lupine invasion in the Veps Forest is escape events from gardens. We have come to a similar conclusion concerning observations in regions further south of European Russia. Namely, in a study of flora of the Trans-Siberian Railway, garden lupine was recovered only on 2 sample plots of 371, and both were located in European Russia (Kostroma and Kirov Oblasts) near permanent settlements. Lupine did not arrive at the rural sites via railway dispersal but rather the opposite, spread onto the railway habitats from surrounding rural territories.

Within protected areas, a strict control of the dispersal of alien species is required, particularly those which show transformative behavior in neighboring geographical regions. In Finland, also near the northern limit of the lupine's range, the species exhibits the highest level of invasiveness and actively intrudes forest margin communities. In 2014, we surveyed invasive populations of the species on a 20 km transect, Helsinki–Esbo: Linnanmäki Park → Kumpula → Alberga → Kilo → Keha → Esbo. High abundance and cover were noted, along with active reproduction by seed. In Finland, 10 variations of corolla coloration were observed in invasive populations (Table 3 and Figure 3). In Russian invasive populations, individuals are represented only by three color forms: violet, pink, and white in the ratio 23:16:7 [22]. In the Veps forest, only the form with the violet corolla is naturalized.

Our conclusion on the high polymorphism of *L. polyphyllus* in Finland is supported by molecular genetic methods [39]. In that study, genetic variability of lupine was determined using 13 microsatellite loci based on samples from 51 sites. The genetic variability of *L. polyphyllus*, i.e., the average number of alleles of a fragment, is not linked with the geographic position of the sites. Invasive populations of *L. polyphyllus* in Finland are genet-

ically diverse, not as a result of an expansion of the lupine, but rather a result of multiple introductions of plants from various sources [39]. The studied invasive populations of garden lupine in Central Russia (Moscow, Smolensk, Ryazan and Kostroma Oblasts, and MBG RAS within the city of Moscow) exhibit interpopulation diversity but lack significant intrapopulation diversity [40].

**Table 3.** Color variability of the flower of *Lupinus polyphyllus* from invasive populations.

| Color Variation | Color of the "Standard" | Color of the "Keel" | Finland | Middle Russia | Veps Forest |
|---|---|---|---|---|---|
| I | violet with rose margin | violet | + | | |
| II | light blue with rose margin | light blue | + | | |
| III | white | white | + | + | |
| IV | dark violet with white margin | dark violet | + | | |
| V | violet | violet | + | + | + |
| VI | violet with purple margin | violet | + | | |
| VII | light rose | rose | + | + | |
| VIII | blue | blue-violet | + | | |
| IX | blue with white margin | blue-violet | + | | |
| X | blue with rose | violet | + | | |

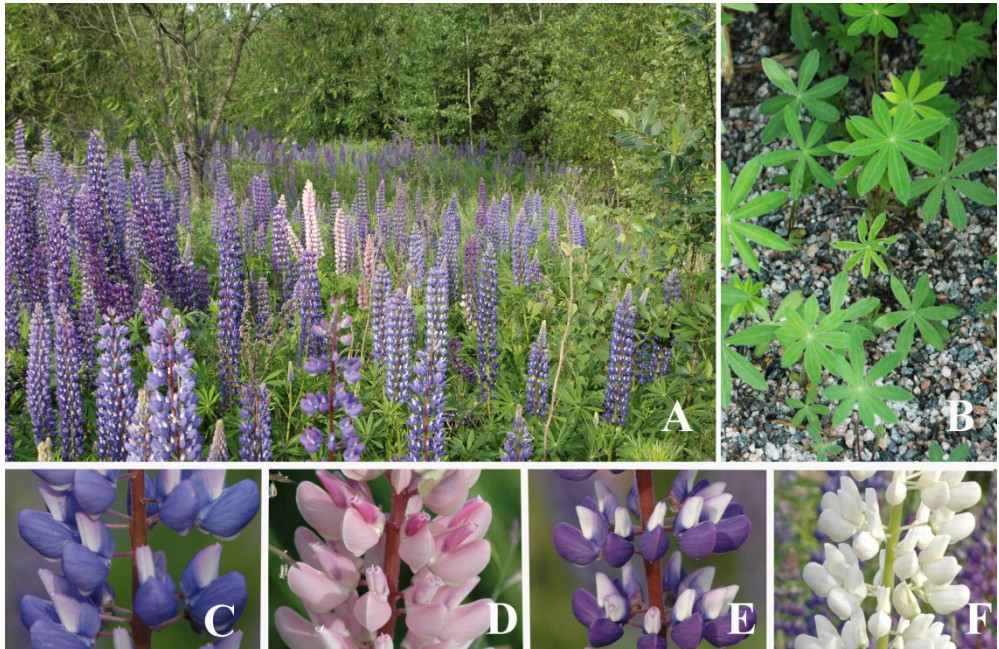

**Figure 3.** Invasion of *Lupinus polyphyllus* in the natural plant communities in Finland. (**A**) Invasive population, 2014, Alberga; (**B**) Mass self-seeding; (**C**–**F**) Color variability of the corolla: (**C**) violet, (**D**) light rose, (**E**) purple, and (**F**) white.

Our studies of *L. polyphyllus* in the Veps Forest revealed a low genetic diversity of the populations. The nuclear ITS fragment was very similar in all samples of garden lupine collected in European Russia. They could be divided among two quite similar haplotypes, one of which is represented by the single specimen LV4c, which belonged to a population from Vinnitsy village (Veps Forest) (Figure 4a). Low ITS variability shows that populations of *L. polyphyllus* in Central Russia (for example, Kaluga Oblast and Tula Oblast) and Northern Russia (St. Petersburg and Veps Forest) have the same origin. Perhaps these populations have common founders that were cultivated in gardens in Central Russia.

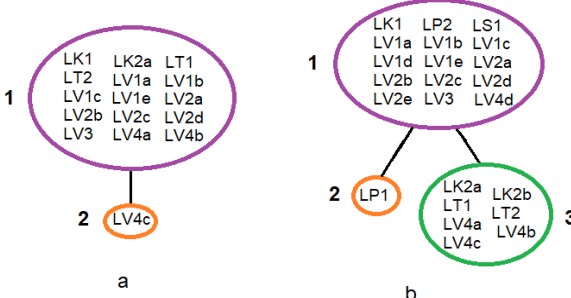

**Figure 4.** Haplotype network of *Lupinus polyphyllus* from different parts of its secondary distribution range: (**a**) Locus ITS; (**b**) Locus rpl32–trnL.

Using SplitsTree and the UPGMA method, we reconstructed a phylogenetic tree based on chloroplast non-coding intergenic spacer rpl32–trnL sequences (Figure 5). A specimen from St. Petersburg was distinct from the others, which in turn could be divided among two clades not correlated with the geographical position of sample collection sites. Samples from the Veps Forest Nature Park and from Kaluga Oblast alike were distributed among both clades. Intrapopulation diversities were shown to be quite significant as well, which suggests the presence of microevolutionary processes near the northern limits of the range of *L. polyphyllus*. This is confirmed by the relatively high bootstrap support (Figure 5).

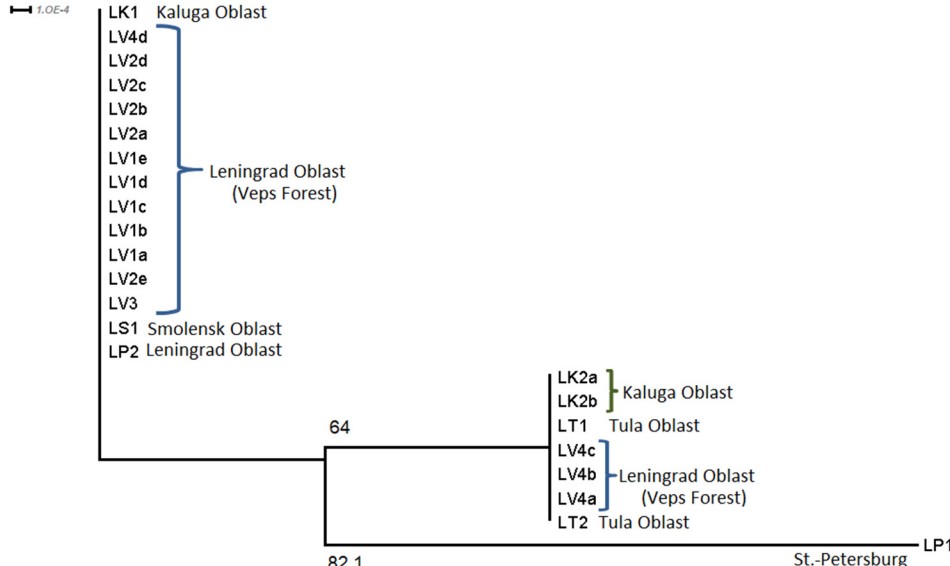

**Figure 5.** The UPGMA tree of *Lupinus polyphyllus* from different parts of its secondary distribution range.

The haplotype network based on the analysis of rpl32–trnL fragments is formed by three haplotypes (Figure 4b) that are very similar to one another. Haplotype 1 is represented by most of the specimens from Leningrad Oblast (most specimens from the Veps Forest and a specimen collected in 1993 without an exact locality specified), one from Smolensk Oblast, and one from Kaluga Oblast; haplotype 2 is represented by a single specimen from St. Petersburg; and haplotype 3 is represented by the remaining specimens from the Veps Forest, two specimens from Kaluga Oblast, and specimens from Tula. The specimen representing haplotype 2 did not fall into either of the two clades reconstructed using the UPGMA analysis.

Genetic analysis of lupine populations was also carried out in Lithuania. Significant genetic differentiation of *L. polyphyllus* populations ($\Phi_{rt} = 0.444$, $p < 0.001$) was demonstrated by ANOVA. However, neither UPGMA cluster analysis nor principal coordinate analysis

revealed a correlation between the results obtained and the geographic locations of the populations [41].

The conducted biochemical analysis revealed a variation of total flavonoid content between 0.86 and 1.63 mg/g dry weight (Table 4) among the samples of leaves. Flavonoids of the quercetin group predominate; they are responsible for the plant's stress tolerance. The polyphenol content revealed low variability; however, in the specimen LV3, collected in a meadow community in Fenkovo village, it reached 7.14 mg/g (Table 4), which was concluded to be caused by biotic stress, most likely related to biomass consumption by herbivores. Note that a high content of flavonoids in lupine leaves was previously found in Central Russia (Smolensk Oblast) [42]. In the habitats indicated in this paper, lupine forms monodominant thickets in Smolensk Region. We obtained comparable high values of flavonoid content, which again indicates a high invasive potential of *L. polyphyllus* in the Veps Forest.

**Table 4.** Total flavonoid and polyphenol content for different samples of *Lupinus polyphyllus*.

| Sample No. | Total Flavonoid Content of the Quercetin Group, mg/g Dry Weight (mean ± SD) | Total Polyphenol Content of the Gallic Acid Group, mg/g Dry Weight (mean ± SD) |
|---|---|---|
| LV1a,b | 1.28 ± 0.34 | 6.34 ± 0.51 |
| LV1c | 1.43 ± 0.31 | 6.82 ± 0.59 |
| LV1d,e | 1.18 ± 0.21 | 5.28 ± 0.45 |
| LV2a | 0.86 ± 0.18 | 4.28 ± 0.39 |
| LV2b,c | 1.34 ± 0.25 | 5.40 ± 0.49 |
| LV2d | 0.99 ± 0.12 | 5.78 ± 0.49 |
| LV2e | 1.09 ± 0.20 | 4.72 ± 0.46 |
| LV3 | 1.63 ± 0.25 | 7.14 ± 0.53 |
| LV4a,b | 1.32 ± 0.14 | 5.40 ± 0.41 |
| LV4c,d | 1.23 ± 0.20 | 4.66 ± 0.38 |

Note: mean—arithmetic mean; SD—standard error of the mean.

Therefore, it may be concluded that the planting material of the garden lupine used near the schools and the museum, as well as the private garden within the Veps Forest Nature Park, has the same origin as in Central European Russia. Notably, the dispersal of *L. polyphyllus* in the Veps Forest takes place not through migration of the species northbound along highways, railways, etc., but rather through escape events from gardens. High phenol and flavonoid content in the leaves of *L. polyphyllus* suggests that the plant has successfully adapted to the new growth conditions. There is evidence that even insignificant cover of lupine reduces the native species' richness, and this negative effect can be detected as early as 5 years after the invasion of the species into the phytocenosis [43]. Therefore, lupine dispersal in this location should be carefully controlled.

## 4. Conclusions

All samples of *L. polyphyllus* grouped into two close haplotypes at the nuclear DNA site and three close haplotypes at the chloroplast DNA site. It has been established that lupine leaves have high phenol and flavonoid content. Flavonoids of the quercetin group predominate.

The obtained data show, on the one hand, a low interpopulation diversity of *L. polyphyllus* within its secondary distribution range, and, on the other hand, a high intrapopulation diversity, which suggests the occurrence of rapid microevolutionary processes in an alien species. Only at the initial stage of an invasion, genetic diversity is reduced within the population of the alien species as a result of founder and bottleneck effects. Over the course of naturalization, the genetic diversity of invasive populations gradually increases. For

this reason, invasive lupine populations in Finland exhibit a higher morphological and genetic diversity compared to the recently escalating populations within the Veps Forest. Furthermore, multiple introduction events may have taken place in Finland, contributing to the rapid differentiation of the populations, while the populations in the Veps Forest all share a single origin of planting material.

**Supplementary Materials:** The following supporting information can be downloaded at: https://www.mdpi.com/article/10.3390/agronomy12102466/s1, Table S1: Samples of *Lupinus polyphyllus*, locations and GenBank accession numbers.

**Author Contributions:** M.A.G.: field research, conceptualization, molecular genetic methods, writing—original draft; Y.K.V.: methodology, data curation, software; V.N.Z.: field research; N.V.V.: molecular genetic methods, software; E.V.T.: writing—original draft; O.V.S.: biochemical analysis, supervision. All authors have read and agreed to the published version of the manuscript.

**Funding:** The work was performed within the framework of the State Task for the Main Botanical Garden of Russian Academy of Sciences, MBG RAS (no. 122042600141-3), State Task for *Library for Natural Sciences RAS* (no. 075-01467-22-00) and agreement no. EP/29-10-21-4 between MBG RAS and Komarov Botanical Institute of Russian Academy of Sciences. The study was supported by the Russian Foundation for Basic Research (grant no. 19-54-26010) and the Ministry of Education and Science of Russian Federation (agreement no. 075-15-2021-1056) and grant no. 075-15-2021-678 for the Centre for Collective Use "Herbarium of MBG RAS", and grant no. FZWG-2021-0018.

**Institutional Review Board Statement:** Not applicable.

**Informed Consent Statement:** Not applicable.

**Data Availability Statement:** Not applicable.

**Acknowledgments:** The authors of this article are very grateful to the researcher of M.V. Lomonosov Moscow State University Dmitry A. Bochkov help to translate this manuscript into English.

**Conflicts of Interest:** The authors declare no conflict of interest.

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
