# Peer review of "Initial Stage of Formation of Spontaneous Invasive Populations of Garden Lupine (Lupinus polyphyllus Lindl.) at the Northern Limit of Its Secondary Distribution Range in the Veps Forest Nature Park"

_agronomy, doi:10.3390/agronomy12102466_

Round 1
Reviewer 1 Report
This study entitled" Initial Stage of Formation of Spontaneous Invasive Populations of Garden Lupine (Lupinus Polyphyllus Lindl.) at the Northern Limit of its Secondary Distribution Range in the Veps Forest Nature Park" detected the specific molecular and biochemical features of the garden lupine at the northern limits of its secondary distribution range. The topic look like well, while the manuscript was poor in logic specially the abstract, introduction and material and methods which needed to be significantly improved.
Comments as below:
1.Abstract: please clearly indicate the objective and experimental design of this study. Meanwhile, add more specific finding or data.
Introduction:
2. Line 39-56: The meaning of these two paragraph is same, better to emerged it or rewrite it.
3. The aim of the study was to evaluate the invasion activity of L. polyphyllus through its molecular-genetic and bio- chemical characteristics but I did't find any information or supporting references about the molecular-genetic and bio- chemical characteristics of the studied species or invasive species. Therefore, it is suggested to rewrite the introduction and the logical of introduction should be improved.
4. Line 94: Add more explanation about route-based method.
5. If possible, add the information of database of species nomenclature.
6. Why authors selected the L. polyphyllus as target species?
7. Line 136: there is lack of information about of analysis of data, add more explanations.
8. Why author choose only total flavonoid and polyphenol content in bio- chemical characteristics to evaluate the invasion? Add this information in introduction part as well as in material and methods section.
9. Line 251-254: The obtained data show, on the one hand, a low interpopulation diversity of L. 251 polyphyllus within its secondary distribution range, but on the other hand, a high intrapopulation diversity, which suggests the occurrence of rapid micro evolutional processes in an alien species. How?? Make it clear.
10. Please also highlighted the novelty and importance of this study in the conclusion.
11. Quality of figures should be improved.
Reviewer 2 Report
The manuscript deals with an essential aspect of the naturalization of alien species. A problem that has been known for centuries, in recent decades has been exacerbated by economic activity and climate change.
The manuscript requires major revision. In the introduction should be added a few sentences about the history of the introduction of lupine in Europe. Methods - Molecular data: there should be information - where the specimens were collected and how many specimens from every locality was used. Likewise for polyphenol and flavonoid. Did the content differ significantly between locations?
Results and discussion should be separated. The results do not provide information on how large the total area Lupinus occupied in the places found, and how numerous were the subpopulations. The description of flower color differentiation from other tests and the photographs should be removed - they are not related to the text. There is no proper discussion.
Figure 1 – the quality of the map is very low
Table 2 – plot size, number of species in the plot, the datum should be added
Figure 2 – lack of the photo author
Few language mistakes e.g. shrubs (not schrubs), in a few places lack of prepositions
Reviewer 3 Report
The manuscript "Initial Stage of Formation of Spontaneous Invasive Populations of Garden Lupine (Lupinus Polyphyllus Lindl.) at the Northern Limit of its Secondary Distribution Range in the Veps Forest Nature Park" is very well written and suitable for publication in Agronomy. However, some corrections are needed.
Title: You should correct the scientific name in the title "Lupinus polyphyllus Lindl."
Abstract
The methodology used was not made clear here.
Introduction
You could write more of the species Lupinus polyphyllus, type survival mechanisms (seed production, dormancy, dispersal, propagation, etc), for a better direction of the reader.
After the objective, you should show how your results will help in the management of this invasive plant.
Materials and Methods
Show the location of the study in Russia, such as geographical coordinates, soil type, climatic conditions, etc.
Line 130: Correct the reference Kuzmenko et al. (2019) [19]
Data analyses: how did you do it? how was the sampling? You should describe it better.
Did you sample the infestation of the invasive plant?
Results and Discussion
Line 141: How many leaves? How long after emergence?
Line 153. What does semi-weedy species mean?
Table 3 and Fig. 3. In Finland? And in Russia?
Fig. 3. The last three pictures have no letters, D, E and F?
Lines 201-206. Letters in a different style from the text.
Table 4. The number of samples is not clear. What does it mean? Are the values the means and +/- the deviation or error from the mean?
The discussion of the results can be improved based on the scientific literature.
Conclusion
You could present the main results.
In the introduction and the objective, it was not shown that you would compare weed populations in Finland and Russia. Rewrite!
References
All are adequate for the manuscript.
Round 2
Reviewer 3 Report
Line 314: Change "AMOVA" to "ANOVA"
Author Response
Dear Reviewer!
Thank you for your review of our article.
We have corrected the error you have noted.
